# *Hyalomma* spp. in Austria—The Tick, the Climate, the Diseases and the Risk for Humans and Animals

**DOI:** 10.3390/microorganisms10091761

**Published:** 2022-08-31

**Authors:** Georg Gerhard Duscher, Stefan Kienberger, Klaus Haslinger, Barbara Holzer, Irene Zimpernik, Reinhard Fuchs, Michael Schwarz, Peter Hufnagl, Peter Schiefer, Friedrich Schmoll

**Affiliations:** 1AGES Research Services, Austrian Agency for Health and Food Safety, Spargelfeldstrasse, 1220 Vienna, Austria; 2ZAMG—Zentralanstalt für Meteorologie und Geodynamik, 1190 Vienna, Austria; 3Division of Animal Health, Austrian Agency for Health and Food Safety, 2340 Mödling, Austria; 4Department for Data, Statistic and Risk Assessment, Austrian Agency for Health and Food Safety, 8010 Graz, Austria; 5Department for Data, Statistic and Risk Assessment, Austrian Agency for Health and Food Safety, 1220 Vienna, Austria; 6Division for Public Health, Austrian Agency for Health and Food Safety, 1090 Vienna, Austria; 7Office of the Provincial Government of Salzburg, Veterinary Directorate, 5020 Salzburg, Austria

**Keywords:** Hyalomma, CCHF, risk, migratory birds

## Abstract

Recently, ticks of *Hyalomma* spp. have been found more often in areas previously lacking this tick species. Due to their important role as a vector of different diseases, such as Crimean-Congo-hemorrhagic fever (CCHF), the occurrence and potential spread of this tick species is of major concern. So far, eight *Hyalomma* sp. ticks were found between 2018 and 2021 in Austria. A serological investigation on antibodies against the CCHF virus in 897 cattle as indicator animals displayed no positive case. During observation of climatic factors, especially in the period from April to September, the year 2018 displayed an extraordinary event in terms of higher temperature and dryness. To estimate the risk for humans to come in contact with *Hyalomma* sp. in Austria, many parameters have to be considered, such as the resting place of birds, availability of large livestock hosts, climate, density of human population, etc.

## 1. Introduction

Since autumn 2018, adult ticks of the *Hyalomma marginatum* species complex have been more frequently found in various countries of central, western and northern Europe [1,2]. Before then, findings of *Hyalomma* sp. in these areas of Europe, e.g., the Czech Republic, Denmark, Finland, Germany, Norway, Poland, the United Kingdom, Slovak Republic, Sweden and the Netherlands [3,4,5,6,7,8] were rather sporadic and rare. Usually, the distribution of ticks of this genus is endemic in the semi-arid zones of African, Asian and Mediterranean countries. Outside of these areas, every spring, the ticks are moved via migratory birds to northern areas along predetermined flight routes of these birds. In the place of origin, the ticks start their spreading mainly via infestation of birds as larvae. As they are two-host ticks, the larvae feed on birds, moult to nymphs and feed again on the same host. After up to 26 days [5,6], the ticks drop off as engorged nymphs anywhere along the flight route of the migratory bird. 

The estimated total number of migrating birds is 2.1 billion per year [9]. 

Several studies from different bird ringing stations provide a picture of the ratio of migrating birds harboring ticks. A study on Ponza Island (IT) from Battisti et al. (2020) reported that a proportion of approximately 11% of these birds carry ticks [10]. In contrast, on 2.7% of the birds, ticks were found in another study [11]. 

However, different special or specific requirements of the climate have to be met to ensure further development of the tick to the adult stage. In areas of permanent *Hyalomma* population occurrence, temperatures exceed a cumulative temperature of 800 °C from September to December (~120 days), whereas in areas with no established permanent population, this temperature stays below 400 °C [12]. Under laboratory conditions, engorged nymphs require about 300 °C cumulatively (constantly above 14–16 °C) to complete moulting. Scarce knowledge is available about the humidity or aridity required for these ticks, although it is stated that, especially for the ticks from southern areas in Africa, (e.g., *H. rufipes*) the aridity represents an important parameter [13,14]. Concerning cold temperatures, it is known that these ticks can survive and hibernate even under harsh conditions at degrees below zero and maintain a stable population over the years until the development of the ticks can be fulfilled in summer and autumn [12]. In the case that the engorged and dropped-off nymphs encounter the required conditions for moulting to adult males and females, they, consequently, can quest or seek for hosts in the new habitat, or even hibernate [12]. Unlike the species of hard ticks that occur autochthonously in these parts of Europe, which quest for hosts while waiting on the grass or bushes, *Hyalomma* spp. ticks actively search for potential hosts. In their usual habitat, adult *Hyalomma* spp. primarily infest cattle and horses, but humans can also be attacked [14]. After biting the hosts, the adults mate, and the dropped-off females lay thousands of eggs, out of which larvae hatch. These larvae infest, e.g., small rodents or birds. To our knowledge, so far, only the development to adults and feeding was observed in northern countries, but not the laying of eggs or hatching of larvae. 

In addition to the aspect that these actively hunting ticks represent an extraordinary and unknown threat, *Hyalomma* spp. ticks are vectors of agents of non-indigenous diseases, some of which are of zoonotic character, such as Crimean-Congo-hemorrhagic fever (CCHF) and rickettsioses. CCHF, especially, is of major concern due its ability to be transmitted via ticks, the blood of infected animals but also via nosocomial route from human to human [15]. CCHF is caused by an *Orthonairovirus*, belonging to the family *Nairoviridae* of the order *Bunyavirales*. This virus is supposed to have a lethality of 5–30%. So far, cases of CCHF or CCHFV in ticks could be found in Kosovo [16] and more recently in Spain [17]. Interestingly, CCHFV could be identified in Hungary in ticks, and seropositivity in hares, rodents and humans has been reported [18]. 

To date, there is no known case in the literature where a tick from migratory birds has been tested positive for CCHFV, but this cannot be ruled out in the future. Furthermore, *Hyalomma* ticks from birds were found positive for *Rickettsia aeschlimannii* and *R. africae*, which also act as zoonotic agents. The prevalence rate of these bacteria is supposed to be 20% to 48% [2,10] of *Hyalomma* spp. collected from birds. 

The most recent IPCC (Intergovernmental Panel on Climate Change) assessment report, focusing on risk management and adaptation, strengthened the need to understand and manage climate-related risks as a multi-dimensional concept integrating drivers of hazard, exposure and (social) vulnerability [19]. The conceptual approach, which has been harmonized in the fourth assessment report [20], links very well to risk management in general, e.g., as understood in the Disaster Risk Reduction community [21], and increasingly also in the context of the public health domain, especially in the context of vector-borne diseases (e.g., [19,22]).

In this context, the ‘One Health’ concept is of greatest relevance. It evolved at the beginning of the 2000s and states that human health is strongly connected to that of animals, the environment and plants, as they continuously interact among each other [23,24,25].

It is therefore of relevance that the management of prevailing and future risks will be embedded in a framework, which considers environmental along with societal drivers. Understanding the management of disease risk as a holistic process allows for a shift from responding to disease prevalence alone to the understanding and the management of disease risk. Therefore, information and data are required on the environment (incl. climate), its human interactions and the risk and vulnerability of society [22].

This paper aims to elucidate the situation in Austria as an example of a country located in the middle of Europe with non-autochthonous occurrences of *Hyalomma* ticks and pathogens transmitted by them:i.Evaluate tick findings;ii.Evaluate and monitor the serological status of CCHF in sentinel animals;iii.Analyse retrospective and prevailing climatic conditions during the tick season in this certain area to estimate the potential for development of *Hyalomma* ticks;iv.Describe potential resting places for migratory birds and densities of hosts (large animals);v.Determine the risk for humans.

## 2. Materials and Methods

### 2.1. Tick Findings

Tick findings depend on citizens’ notifications. After the accidental finding in 2018 [1], the public was informed during the annual press conference and press release of some current information on tick-borne encephalitis. People were asked to report any suspicious tick findings by sending a photographic image to the author. After confirmation of the *Hyalomma* spp. in the photo, transport was organized and analysis of the tick was performed. Tick determination was performed according to morphological keys [14], and pathogen detection was conducted by using diagnostic PCR methods (High Pure Viral Nucleic Acid Nucleic IsolationKit (Roche Diagnostics GmbH, Mannheim, Germany)). PCRs were processed using commercial kits (RealStar CCHFV RT-PCR Kit 1.0 (Altona Diagnostics, Hamburg, Germany)) and protocols targeting the ITS (internal transcribed spacer) of *Rickettsia* sp. and 18 S rRNA of Piroplasmidae, as previously described [1].

### 2.2. Serology on CCHF Antibodies

In 2018, sera of 897 cattle all over Austria were collected according to a sampling plan based on cattle density (Figure 1). Only cattle with a minimum of a 2-month grazing period prior to sampling have been included. The sera were analysed performing an antibody ELISA test (ID Screen^®^ CCHF Double Antigen Multi-species from IDVet (Grabels, France)). A double antigen ELISA for the detection of antibodies against Crimean-Congo haemorrhagic fever virus (CCHFV) can be conducted in serum, plasma and blood filter paper samples from cattle, sheep, goats or other susceptible species. The test is valid when the mean value of the optical density of the positive control (ODPC) is bigger than 0.350 and the ratio of the mean values of the positive and negative controls (ODPC and ODNC) is above 3 [26]. Furthermore, for each sample, the sample to positive (S/P) percentage (S/P%) was calculated: S/P% = ODSample/ODPC x100, where an S/P% below or equal to 30 is negative and an S/P% higher than 30 is considered positive. 

### 2.3. Climate

For analysing the retrospective data in the context of climatic conditions, we conducted gridded climate observations for the Austrian domain. Specifically, we used the SPARTACUS data set [27,28], which consists of daily fields of maximum and minimum temperature as well as daily precipitation sums. The spatial resolution is 1 km × 1 km, and the temporal coverage is from 1961 to yesterday (SPARTACUS provides operational climate monitoring). Daily mean temperatures were approximated via (Tmax + Tmin)/2. For assessing the temporal evolution during the respective year, daily mean temperatures were averaged over the whole domain of Austria and accumulated over the period from April to September. In addition, the annual anomalies of the temperature sums from the 1st of April to the 30th of September with respect to the reference period of 1961–2020 were given. This period of the year was chosen because it is the crucial time span for the *Hyalomma* ticks to arrive with the birds and to develop further to adults. As a drought indicator, we used the Standardized Precipitation Evapotranspiration Index (SPEI, [29,30]). The SPEI is calculated by transforming the probability distribution of the climatic water balance into a unit normal distribution. The climatic water balance is given by the difference between precipitation and atmospheric evaporative demand (AED) over a given time period. This enables an intuitive assessment of dry (SPEI negative) and wet (SPEI positive) conditions [31,32]. Here, we used the SPARTACUS precipitation sums and AED data from Haslinger and Bartsch (2016), which consists of daily AED values on the same spatial grid as SPARTACUS [33]. The SPEI was calculated on a 30-day accumulation window and then averaged from April to September.

### 2.4. Migratory Birds, Available Hosts

A map of Austria’s avian influenza (AI) risk areas is used as an approximation for the most likely occurrence of migratory birds [34]. The risk areas on this map were defined based on various parameters. Primarily, communities along large bodies of water and with positive AI findings in the past were selected; additional areas were evaluated based on expert knowledge. 

Livestock information for cattle and horses was obtained from the Austrian animal husbandry database (Verbrauchergesundheitsinformationssystem, VIS), which covers the year 2018. The database includes the geographical location of each farm. This information was used to aggregate the number of animals into 5 km grids (regional statistical grids—data.statistik.gv.at—under consideration of data protection). Horses are covered by the database only in the case of a reporting obligation because of other livestock animals (cattle, pigs, small ruminants); this leads to an underestimation of the horse population (compare Pferd Austria, IWI, 2019). 

### 2.5. Risk

Based on the conceptual definition of risk in the context of climate change, we followed the approach of climate impact chains to identify causal drivers of risks for an infective bite by *Hyalomma* sp. and a pathogen. Climate impact chains are an analytical concept to better understand, systemize and prioritize the climate, environmental and socio-economic drivers of related hazards, vulnerabilities, exposures and risks in a specific system [35].

We applied an integrated conceptual health risk and vulnerability framework which (i) considers the notion of multiple inter-related factors contributing to disease risk; (ii) provides a clear framing of risk and vulnerability in-line with current IPCC recommendations; (iii) establishes a clear link to risk governance, climate change adaptation and related intervention measures; (iv) allows for the identification of possible development pathways; and, finally, (v) provides a holistic view of disease risk considering spatial and temporal scales. This approach supports the identification and interrelationships of risk drivers within a systemic view. 

## 3. Results

### 3.1. Tick Findings

So far, eight ticks (Figure 2) were reported via photographic images (Figure 3) and were explicitly identified as *Hyalomma* sp. Out of these, three could be further processed, whereas the others were disposed prior to exact species determination and pathogen analysis (Table 1).

### 3.2. Serology in Cattle

None of the 897 sera delivered a positive result for CCHF antibodies. For ODNC and ODPC, values of 0.04 and 1.0, respectively, were found. So, all validation criteria for the ELISA test were fulfilled. All serum samples reached an S/P% between 3 to 14%, therefore, they were clear negative samples. 

### 3.3. Climate 

Starting from 1961 and in the observed timeframe from April to September, 2018 displayed an extraordinary event if looking at accumulated daily temperatures (Figure 4). Concerning the dryness (SPEI—Standardized Precipitation Evapotranspiration Index), April until September 2018 is recognized as dryer than normal (Figure 5).

### 3.4. Migratory Birds, Available Hosts

The places with an assumed higher density of migratory birds are derived from the AI risk map (Figure 6) [34].

Slightly less than two million cattle are kept in Austria, around 50% each in Upper and Lower Austria. Around 75,000 horses have been reported in the VIS database for 2018; approx. 40% are kept in Upper and Lower Austria. The geographical distribution of cattle and horses in Austria is unequally distributed (Figure 7).

### 3.5. Risk

Impact chains can serve as a heuristic—and often expert-based—approach to identify drivers and ultimate indicators for quantitative and qualitative climate risk assessments (Figure 8).

For the risk of an infective bite by *Hyalomma* sp., an important causal hazard driver is the changing landscape, which causes increased habitat and survival options for *Hyalomma* sp. For Austria, this is specifically based on higher temperatures and longer dry periods (see also section on climate). Together with the presence of a pathogen and changing patterns of migrating birds and related locations for bird ‘stop-overs’, this may lead to an increased probability of the presence of an infected *Hyalomma* sp. Relevant factors in terms of human risk include the presence of livestock at farms as possible hosts (Figure 7) and general exposure of the human population to these ticks (Figure 9). The latter could also be stratified—depending on the relevant pathogen—by age and sex. Highly relevant is the exposure of specific population groups due to outdoor activities, especially together with livestock, such as horse riding and farming. Drivers of vulnerability, and therefore also relevant resilience factors including the accessibility to health services, as well as awareness of ticks and the related behaviour to avoid bites, are also important. The complex interaction of these different drivers is likely responsible for an increase or decrease in the risk of coming in contact with *Hyalomma* sp. ticks bearing an infective agent.

## 4. Discussion

### 4.1. Tick Findings

So far, the findings of *Hyalomma* ticks in Austria is still sporadic and most probably strongly biased in terms of different awareness in the public. In regard to this data, it is not possible to derive reliable distribution and occurrence data of the ticks. However, their finding places might reflect areas where the climatic situation favoured further development and where there are areas of higher concern in terms of new *Hyalomma* tick hot spots. 

### 4.2. Serological Data

Cattle represent the preferred hosts for adult *Hyalomma* ticks and develop an immune response to CCHFV. Therefore, they can be taken as indicator animals. Although no positive cattle were identified in this study, the results are of great importance while observing the epidemiological situation. These data represent the baseline and can be compared to future monitoring studies. 

### 4.3. Climate

In the year 2018, which was the year where *Hyalomma* ticks were found in a lot of northern European countries, vast areas across Europe experienced significant above-normal temperatures during the summer season. For Austria in particular, accumulated temperatures from 1 April to 30 September were constantly above all other years on record from 1961 onwards (Figure 4a) and reached an extraordinary peak in the temperature sum anomaly, as depicted in Figure 4b. Through distribution fitting following a bootstrapping approach, the return period of the year 2018 is an estimated 358 years with a standard error of 79 years, highlighting this significant anomaly. The effects of global climate change are clearly visible from the apparent increase in the warm-season temperature sum anomaly, which shows a steep increase from roughly 1980 to 2010, in line with the general temperature increase in Austria and the Alpine Region [36]. This observed trend clearly favours the development of engorged nymphs to adults. 

Precipitation: Until now, the importance of dryness was not fully understood in terms of the development of *Hyalomma* ticks. It is assumed that tick species originating from the more southern areas (*H. rufipes*) are more prone to higher humidity. In any case, during breeding experiments on a Mediterranean tick *Rhipicephalus sanguineus,* too high humidity was found to lead to deformations of the ticks during moulting (not published, lit). The year 2018 was a dry year (Figure 5), depicting the warm season SPEI, but there is not such a clear trend as seen with the temperature. Furthermore, very dry conditions prevailed during 2003, but the accumulated temperature anomaly was not as striking as in 2018. Although the occurrence of *Hyalomma* in this year cannot be ruled out, since the tick findings are very scarce, the likelihood that 2018 represents the better year for tick development is rather obvious. 

Most probably, the combination of temperature and humidity, with a higher emphasis on temperature, is necessary. 

### 4.4. Migratory Birds, Available Hosts

Since migratory birds are the major drivers in spreading the ticks, we concentrated on areas where a higher density of birds is known to occur. As an approximation, the data from avian influenza (AI) monitoring areas were chosen. These data are strongly linked to waterfowls, watercourse and the availability of water, which is also necessary for the resting of migrating birds. We assume that these areas are also areas where migratory birds tend to rest and stay longer, increasing the likelihood of dropping engorged *Hyalomma* nymphs. Of course, this is a rough estimation, but with raised awareness and further cooperation, these maps should gain more accurateness. 

Once the ticks have arrived via the birds and the climate conditions are convenient for the ticks to further develop, the ticks either might seek for sheltered places to overwinter or try to find hosts to feed on. In the latter case, large animals such as cattle, horses but also goat and sheep act as preferred hosts for the adult ticks. Therefore, we queried data from the national database VIS to analyse the density of these animals to estimate the possibility of ticks to find a host. We assume that the density of potential large livestock hosts is another major driver in the establishment of new populations of *Hyalomma* spp., since the availability of hosts is relevant for ticks. It is rather obvious that there are geographical overlaps in the areas of assumed higher density of migratory birds and a higher density of livestock, such as, e.g., Upper and Lower Austria (north) and Styria/Carinthia (southeast). Until now, *Hyalomma* spp. ticks have been found exclusively in the northern federal states of Austria, but their presence in other regions cannot be ruled out. While raising awareness in the whole of Austria, we hope to obtain more tick findings to obtain a better picture of the distribution of these ticks. In the future, this information (presence of migratory birds and hosts) should be taken into account (e.g., monitoring programmes).

### 4.5. Risk

Taking this together, the map used for risk analysis for avian influence might reflect areas where birds are more likely to occur, which raises the likelihood to drop off engorged nymphs. Once the climatic condition is favourable, as we especially experienced in 2018, ticks develop further to adults. In certain areas, the chance to find livestock hosts is higher due to a higher number of reared and kept animals. Additionally, in these areas, the human population is also higher. All these factors combined lead to areas of a higher risk of coming in contact with *Hyalomma* ticks and raises the chance to become infected with CCHF. Furthermore, it is also likely that new populations will emerge in these locations as migratory birds introduce more ticks, making it more likely that they will find a mate.

The risk assessment, building on a first version of an impact chain, highlights the interconnected, systemic and multi-dimensional nature of the risk of an infective bite by *Hyalomma* sp. The main purpose of risk assessments is to inform risk management with evidence as well as condensed information. So far, we have presented evidence on single key risk driver, which can serve as a starting point to understand related risk patterns. However, to move ahead, a much more integrated approach as well as a data-driven approach would be required, which considers relationships and is able to map and represents hotspots of risk (e.g., following Kienberger and Hagenlocher 2014 [22]).

## 5. Conclusions

To evaluate the risk for humans to come in contact with a *Hyalomma* tick or with pathogens (e.g., CCHFv) in the tick, it is necessary to consider many different parameters. In this paper, we suggested the major driving factors and discussed their specific impact on the occurrence of *Hyalomma* as well as their influence on the establishment of new endemic populations in Austria. The findings in 2018 indicated the survival and development of ticks after their transportation via migratory birds. The preferred resting places of the migratory birds should be considered in the future, although citizen science, such as observation of “bird life”, can also deliver important information on the routes of migratory birds in future. After arrival, the climatic conditions are key factors in regard to the moulting of engorged nymphs to adults. As we saw in 2018, the temperature is of major concern and was an extraordinary event so far. However, climate change in the future can lead to similar events more often. In addition to the climatic conditions, the availability of livestock hosts and large animals is important to deliver hosts for adults to feed on and to establish and maintain new populations. Humans are accidental hosts but are vulnerable to pathogens from the ticks. Information on the factors for overwintering is available for *Hyalomma* sp. ticks in Central Europe, although it is known that once moulted to adults, their resilience to minus degrees increases [12]. 

## Figures and Tables

**Figure 1 microorganisms-10-01761-f001:**
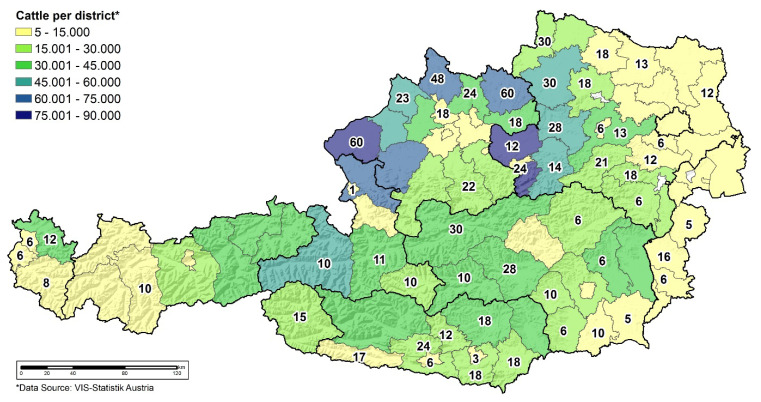
Sampling size per district (numbers) based on the cattle density (colours).

**Figure 2 microorganisms-10-01761-f002:**
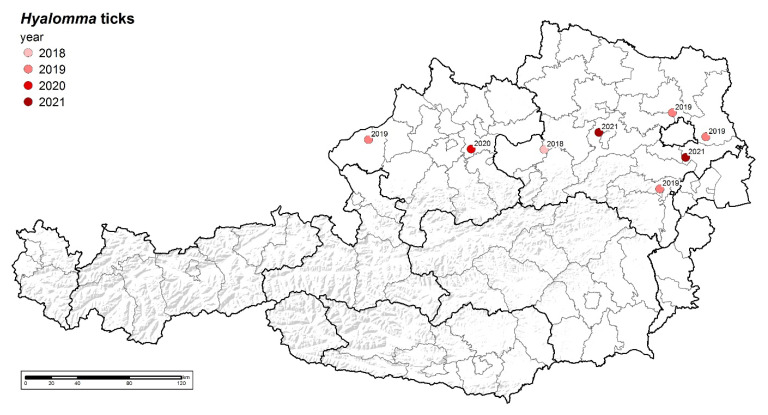
*Hyalomma* sp. findings in Austria in the years 2018–2021.

**Figure 3 microorganisms-10-01761-f003:**
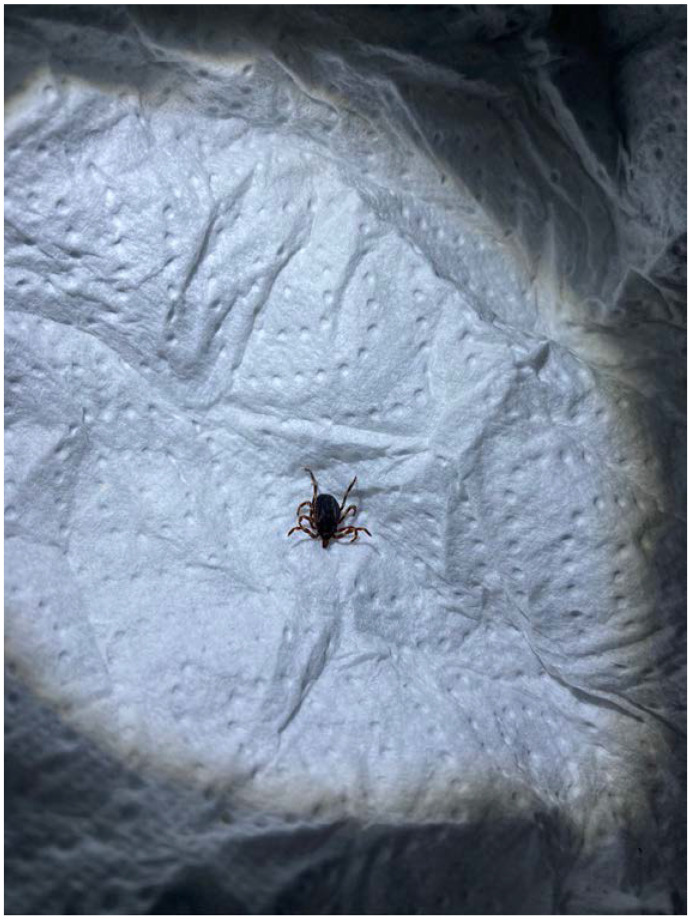
Example of picture sent via citizen science for further determination.

**Figure 4 microorganisms-10-01761-f004:**
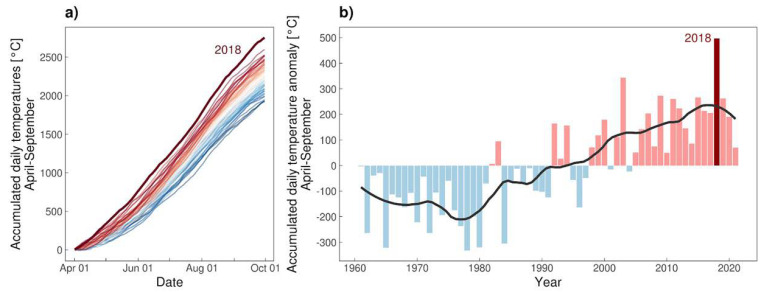
(**a**) Temporal evolution of accumulated daily temperatures (averaged over the Austrian domain) from April 1st to September 30th for each year from 1961 to 2021; the year 2018 is highlighted in darker red. (**b**) Accumulated daily temperature anomalies from April 1st to September 30th from 1961 to 2021 relative to the 1961 to 2020 mean; the year 2018 is highlighted in darker red.

**Figure 5 microorganisms-10-01761-f005:**
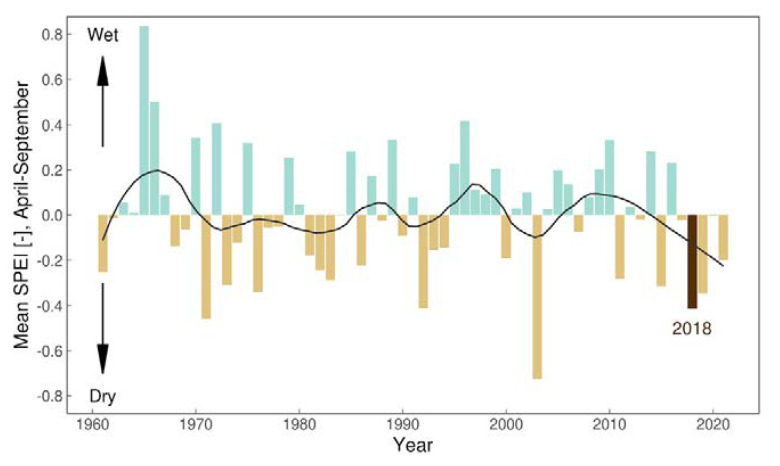
Annual averages of the SPEI (Standardized Precipitation Evapotranspiration Index) for a 30-day accumulation window) for the period April 1st to September 30th. Negative (positive) values denote dryer (wetter) than normal conditions; the year 2018 is highlighted in brown.

**Figure 6 microorganisms-10-01761-f006:**
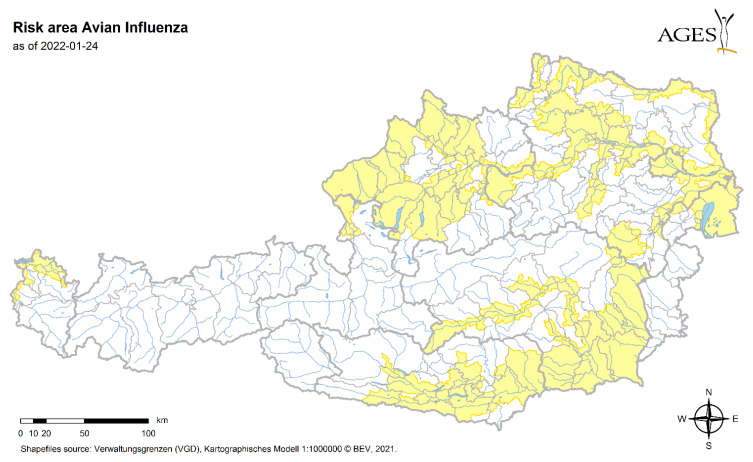
Risk map for Avian Influenza [34]. Areas of assumed higher risk are indicated in yellow. These areas are defined on various parameters such as water bodies, occurrence of birds and previously detected AI. (shapefile source: Verwaltungsgrenzen (VGD), Kartographisches Modell 1: 1000000 ©BEV, 2021).

**Figure 7 microorganisms-10-01761-f007:**
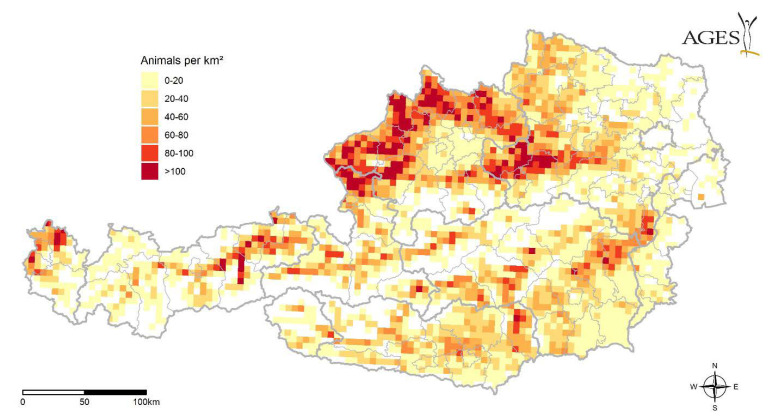
Large livestock animals (horses and cattle) per km^2^. (shapefile source: Verwaltungsgrenzen (VGD) ©BEV, 2021. Regional statictical grid, Statistics Austria. Data.statistik.gv.at, accessed on 1 April 2022).

**Figure 8 microorganisms-10-01761-f008:**
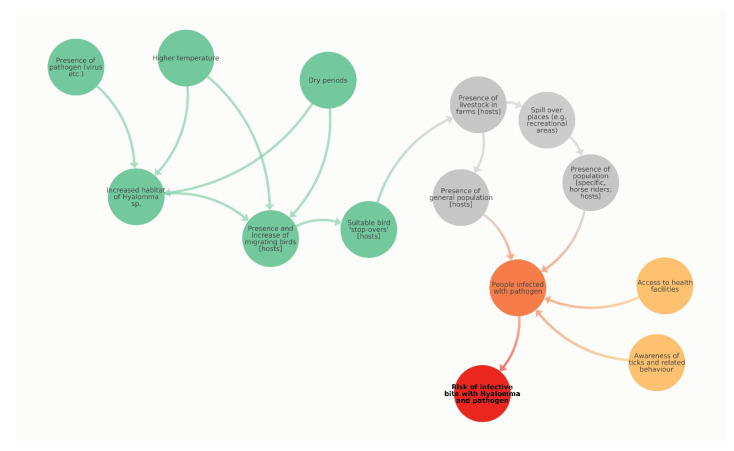
This figure presents a climate impact chain for the risk of an infective bite with *Hyalomma* and a related pathogen. It presents key driving factors and characterizes them according to the risk domain (hazard = green, exposure = grey, vulnerability = orange, intermediate impacts = dark orange, risk = red).

**Figure 9 microorganisms-10-01761-f009:**
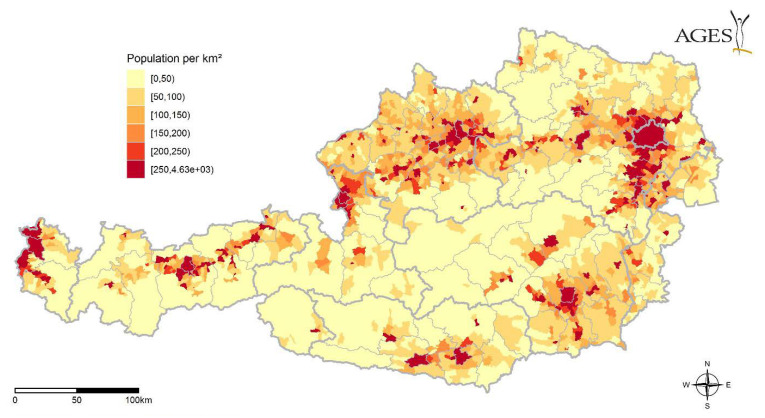
Population of humans per km^2^. (shapefile source: Verwaltungsgrenzen (VGD) ©BEV, 2021. Population data: ©Statistics Austria. Statistics of the population. Created on 17.6.2021).

**Table 1 microorganisms-10-01761-t001:** Species, sex and species of the host were identified in the case of ticks. Microorganisms were detected in the ticks. (CCHFV = Crimean-Congo-haemorrhagic fever virus, na = not available, neg = negative, pos = positive).

Year	Date	District	Species	Sex	Host	CCHFV	*Rickettsia* sp.	*Babesia* sp.
2018	02.10.	Melk	*H. marginatum*	male	horse	neg	pos [1]	neg
2019	26.04.	Braunau	*Hyalomma* sp.	female	human	na	na	na
2019	15.06.	Korneuburg	*Hyalomma* sp.	male	human	na	na	na
2019	28.08.	Wiener Neustadt	*Hyalomma* sp.	na	horse	neg	na	na
2019	05.10.	Gänserndorf	*Hyalomma* sp.	male	horse	na	na	na
2020	03.09.	Linz Land	*Hyalomma marginatum*	male	horse	neg	na	na
2021	30.06.	Bruck/Leitha	*Hyalomma* sp.	na	horse	na	na	na
2021	25.07.	St. Pölten Land	*Hyalomma* sp.	female	horse	na	na	na

## Data Availability

The data presented in this study are available on request from the corresponding author. The data are not publicly available due to different agreements among different partners.

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
