# Peer review of "Hyalomma* spp. in Austria—The Tick, the Climate, the Diseases and the Risk for Humans and Animals"

_microorganisms, 2022, doi:10.3390/microorganisms10091761_

Round 1

Reviewer 1 Report

The manuscript of Duscher et al. is a report of the occurrence of Hyalomma sp. ticks in Austria. The records of Hyalomma ticks were collected by citizen participatory science approach, part of the ticks were identified to species probably by morphological as well as molecular tools. Furthermore, the author tested the ticks for the presence of Crimean-Congo hemorrhagic fever virus, Rickettsia sp., and Babesia sp. and almost 900 serum samples from cattle for anti-CCHFV antibodies. A single tick was positive for Rickettsia (rickettsial DNA?), and no reactive sera was found. Moreover, the authors present a risk assessment model based on an analysis of climatic factors, distribution of migratory bird resting places (approximated from an avian influenza virus risk map), and geographic distribution of livestock in Austria.

The manuscript is in general well written, although mainly the Introduction section needs some improvement as it is quite ungainly, in some parts hard to comprehend. The methods section is well and comprehensibly described. Although I would suggest adding at least some general information on the analysis of ticks (were the species identified morphologically, or also confirmed by molecular biology tools?, which gene target was used), not only a reference to previous research of the authors (which is also quite spare in the methods section). I would also suggest moving section 2.5 to the introduction.

From my point of view, the attempt to provide some general risk assessment framework for the whole topic is the weakest point of the manuscript. I would suggest either skipping the climate impact chain theory and still improving substantially the connections between the different „sections“ of the study – tick records, climate analysis, host (cattle) analysis to get a more complex and comprehensible picture. Also, maybe just „fusion“ of the map outputs – indicating „cumulative“ risk of exposure + records of the actual tick occurrence would help? The climate impact chain seems very general, theoretical with no real connection to the presented data. Also I am quite surprised, that the authors use a term „infective bite“ in section 3.5 as the data are mostly associated with tick presence and survival. Either the risk management should be implemented on tick presence/absence/survival or there should be an attempt to quantify the risk, that the transported tick will be  infected (not mentioning that infected does not necessarily mean infective). Maybe I would prefer to present just the data from tick records, tick analysis (possibly in more detail – how were ticks identified, was the Rickettsia sp. positive sample further characterized?), serology + the climate data as support for the hypothesis, that occurrence of Hyalomma sp. may be driven by climatic changes + a map showing overlap of high cattle density, high human density, and records of ticks? Using the AIV risk map as a proxy for the frequency of migratory bird occurrence seems quite hard to be justified from my point of view.

The discussion is again quite far from the actual results of the study and very general. Moreover, some important points are missed: Which developmental stages of Hyalomma spp. ticks are likely to be transferred by the migratory birds? Are there some alternative paths? Are the Hyalomma spp. ticks able to overwinter? What are the prevalence rates of the tested microorganism in the ticks in areas of their endemic occurrence, are the areas of CCHFV circulation associated with the migratory routes of birds to Central Europe – indicating the risk that the transferred tick will be infected.

Specific comments:

Abstract:

Line 15: I would suggest „Recently Hyalomma spp.“ => „Recently ticks of Hyalomma spp.“

Line 16: Please, explain CCHF abbreviation before the first use.

Line 17: spread of this these tick species => spread of this tick species

Line 18: CCHF antibodies => anti-CCHF (or antibodies against CCHF) + probably CCHF virus more suitable in this context

Introduction:

Line 27: ticks of the genus Hyalomma marginatum complex => ticks of the Hyalomma marginatum species complex?

Line 28: middle Europe => central Europe? And western Europe, as you further mention the UK and the Netherlands?

Line 28: Before that, finds of this genus in northern areas of Europe e.g. the Czech Republic, Denmark, Finland, Germany, Norway, Poland, United Kingdom, Slovak Republic and Sweden and the Netherlands [3-8] were rather sporadic and rare.

=> Before that, findings or records? of these species? or of Hyalomma sp.? in these areas e.g. the Czech Republic, Denmark, Finland, Germany, Norway, Poland, United Kingdom, Slovak Republic, and Sweden, and the Netherlands [3-8] were rather sporadic and rare.

Line 32: Usually the distribution of this tick genus is => Usually the distribution of  ticks of this genus/or species? is (it is the ticks that are distributed not the genus)

Line 40: deliver => provide?

Line 42: delivered => reported?

Line 44: However, different special or specific requirements on the climate have to be met is needed to ensure further development of the tick to the adult stage.

Line 45: in areas of permanent Hyalomma population => in areas of permanent Hyalomma population occurrence

Line 54: Sorry, I am not able to understand this part: „until the development degree is fulfilled in summer and autumn“ => until the development is completed?

Line 55: In case the dropped of nymphs encounter this prevailing and required conditions a moulting to adult males and females can occur, which consequently can hunt for hosts in the new habitat or even hibernate => In case the engorged and dropped off nymphs encounter the required conditions moulting to adult males and females can occur, which consequently can quest or seek for hosts in the new habitat or even hibernate

Line 57: Other than autochthonous ticks, which request hosts while waiting on the grass or bush, this genus is actively searching for potential hosts. => Unlike the species of hard ticks that occurred autochthonously in these parts of Europe, which quest for host while waiting on the grass or bush, Hyalomma spp. ticks  are actively searching for potential hosts.

Line 65: active hunting ticks => actively hunting ticks

Line 66: Hyalomma spp. are vectors of non-indigenous diseases some of which of zoonotic character like Crimean-Congo-hemorrhagic fever (CCHF) and rickettsiosis… => Hyalomma spp. ticks are vectors of cuative agents of non-indigenous diseases some of which are of zoonotic character like Crimean-Congo-hemorrhagic fever (CCHF) and rickettsioses.

Line 70: Please, be more specific with the taxonomic classification of CCHFV. What is meant by „caused by Bunyavirus?“ member of the family Bunyavirales?Its family Nairoviridae,  genus Orthonairovirus; https://talk.ictvonline.org/taxonomy/

Line 77: The proportion … of Hyalomma spp.=> The prevalence rate … in Hyalomma spp.

Line 82: finding => findings

Line 84: Sorry, I am not able to understand aim III., please rephrase.

Line 85: densities of hosts

Line 90: I believe, the authors meant press conference and press release of some current information on tick-borne encephalitis, nevertheless „release ob tick-borne encephalitis may sound a bit misleading?

Line 94: Please, add at least a very brief description of the method to the reference.

Methods:

Section 2.5 – transfer as a last paragraph of Introduction?

Section 2.2 – is there any additional information to share? Like detection limit, cut-off value for positive (borderline) samples? As all the samples were negative, it would be nice to show the method worked properly and is sensitive enough.

Results:

Line 167: extra full stop at the end of the sentence

Table 1: Please rephrase the caption: “Species, sex, host….found in the ticks found“. Species, sex and specie soft he host were identified in the case of ticks. Pathogens were detected in the tick. Actually, there was only Rickettsia  sp. detected in the samples, and there is no indication that it was pathogenic to human or animals – use microorganisms instead? Morover, CCHF – is the disease, should be CCHFV and probably Rickettsia  sp., Babesia sp. Human is with capital H whereas horse is not. In the case of unknown species of the tick, it should be probably Hyalomma sp. there is a missing value for sex in the sample from 28.8.2019. Is it necessary to include the year and repeat it in the date?

Figure 4: Is there some statistical method to evaluate the statistical significance of the „extraordinarity of 2018 temperatures? some kind of “outlier” detection?

Line 199: The geographical distribution of cattle per area unit is unequal, not uniform?

Line 210: ultimately indicators => ultimate indicators

Section 3.5 belongs to Methods, or the totally general  parts even Introduction?

Figure 8: Low quality at least in the .pdf version of the manuscript.

Line 223: „and the general exposure of the population“ => and the general exposure of the human population? It still does not make much sense: „Relevant factors for exposure include and the general exposure of the human population?“

Line 226: horse riders and farmers => horse riding and farming (activities)

Line 229: Sorry, I am not able to understand the last sentence of the paragraph, please rephrase.

Line 376: influenza

Line 275: Are the species of highly infested migratory birds really associated with the water bodies as their resting places? Is there some reference available? Definitely there are some data form trapping of migratory birds on their way to Europe.

Line 284: Are there relevant data that would indicate the possibility of the establishment of independent populations of Hyalomma spp. ticks?

Line 298: ?quit CCHF?

Line 324: „humans are accidentally hosts“ => „humans are accidental hosts“

Line 326: Reference missing?

Reviewer 2 Report

I have a couple of major issues with this manuscript:

1. Major mistakes such as 400  Celsisus degrees (see attached) plus the English language style and grammar needs to be revised. Some sentences just seem to be added on without any connection to the prior content.

2. I don't think you have sufficient data with three human samples, all the three blood samples came back negative for CCHF but one was positive for rickettsia. Rickettsia are very common in ticks but most of them are not pathogenic to humans so this is even not useful. Overall, I think the human data should be NOT part of your manuscript.

3. I like the idea of CCHF sentinel surveillance in livestock as an alert system especially in the times of climate change. This would be not only useful for agriculture but also Public Health. Maybe expand on this to discuss the pros and cons or a cost benefit analysis?

Also, collect more data on Hyalomma ticks to batch collected ticks and test them for CCHF and human pathogenic rickettsia. Not sure if this is premature but was commonly done during the first West Nile Virus (WNV) epidemic in the US in 2002 where mosquitoes, wild birds, sentinel chicken and horses were tested as alerts for potential human WNV disease. Maybe then also submit to a different journal with a different reader population? I would think this should be relevant to agriculture and veterinarians?

Attached some of my revisions, as you see I stopped reveiwing specifically fairly early on.

Round 2

Reviewer 1 Report

Manuscript ID: microorganisms-1849122

Hyalomma spp. in Austria – The tick, the climate, the diseases and the risk for humans and animals

Georg Gerhard Duscher * , Stefan Kienberger , Klaus Haslinger , Barbara Holzer , Irene Zimpernik , Reinhard Fuchs , Michael Schwarz , Peter Hufnagl , Peter Schiefer , Friedrich Schmoll

I would like to thank the authors for taking my comments in consideration. I agree with the majority of the corrections (apart from a few additional minor changes suggested bellow and associated mainly with the new/revise sections). I strongly agree that monitoring and reporting of potentially invasive ticks species (especially vectors of such serious pathogens as CCDFV) is of high importance. Also, the intention to present the finding in a larger conceptual picture of a multifactorial risk assessment is interesting. The authors present nicely the input data from multiple sources, but I still miss the integration of the data into an single indicator of the infection/infestation risk. In short I am convinced my objections summarized in point 5. of the original review are still valid also for the revised version of the manuscript. Nevertheless, it is probably on the opinion of the editor, whether lack of the suggested integrative step/output is a key feature preventing  acceptance of the manuscript for publication or not. Otherwise, I think the manuscript was improved substantially (although some parts seem still quite “heavy footed” language-wise) and may be accepted after the suggested minor corrections.

Specific comments:

Line 64: that occurred => occur

Line 113: I am sorry, I still don´t understand what do you mean by “prevailing” – current climatic conditions prevailing geographically in the whole country? Or during the year or during the ticks season?

Line 139: 0,350 => 0.350

Line 140: I would suggest to delete (ODPC >0.35) as it brings no new information.

Line 141: I would suggest to delete (ODPC and ODNC) as it brings no new information.

Line 143: counted => considered

Line 222: Out of these three could be further processed => Out of these, three could be further processed?

Line 235: Elisa => ELISA

Line 235: therefor => therefore

Figure 6 caption: I guess, that the yellow colour indicates the risk areas, nevertheless, the caption could be more detailed? Including some general description how were the risk areas defined?

Figure 8: the writing in the figure still hard to read – at least in the PDF copy provided for review.

Line 281: general exposure by these ticks of the human population => general exposure of the human population to these ticks

Line 287: are you sure about “draw responsible”? => “are responsible” “are likely responsible”?

Line 289: Hyalomma sp. bearing an infective agent. => Hyalomma sp. ticks bearing an infective agent.

Line 340: cooperationscoperations??

Line 377: we identified the major driving factors => we suggested  the major driving factors – as there were no causal links proven between the tick findings and the described factors

Line 390: Scarce information on the factors for overwintering is available => Scarce information on the factors for overwintering is available for Hyalomma sp. ticks in Central Europe

Discussion:

Is there a reason why there are no findings of Hyalomma sp. ticks in the southern part of the country where avian influenza risk areas and areas of increased density of cattle farming overlap?

Reviewer 2 Report

Thanks for responding to my comments and revising some paragraphs based on my feedback. Your manuscript  just needs some minor text editing (by you and/or the editors) and then should be ready to be published.

Congratulations!

Author Response

Thank you for your comments.